# Inducible expression of large gRNA arrays for multiplexed CRISPRai applications

William M. Shaw[1,2,3], Lucie Studená [1,2,3], Kyler Roy[1,2], Piotr Hapeta[1,2], Nicholas S. McCarty[1,2], Alicia E. Graham[1,2], Tom Ellis [1,2] & Rodrigo Ledesma-Amaro [1,2]

CRISPR gene activation and inhibition (CRISPRai) has become a powerful synthetic tool for influencing the expression of native genes for foundational studies, cellular reprograming, and metabolic engineering. Here we develop a method for near leak-free, inducible expression of a polycistronic array containing up to 24 gRNAs from two orthogonal CRISPR/Cas systems to increase CRISPRai multiplexing capacity and target gene flexibility. To achieve strong inducibility, we create a technology to silence gRNA expression within the array in the absence of the inducer, since we found that long gRNA arrays for CRISPRai can express themselves even without promoter. Using this method, we create a highly tuned and easy-to-use CRISPRai toolkit in the industrially relevant yeast, *Saccharomyces cerevisiae*, establishing the first system to combine simultaneous activation and repression, large multiplexing capacity, and inducibility. We demonstrate this toolkit by targeting 11 genes in central metabolism in a single transformation, achieving a 45-fold increase in succinic acid, which could be precisely controlled in an inducible manner. Our method offers a highly effective way to regulate genes and rewire metabolism in yeast, with principles of gRNA array construction and inducibility that should extend to other chassis organisms.

The capacity of cells to perform complex behaviours is a consequence of a regulated control of the expression of genes. Methods to control gene expression at will, including activation and repression, allow us to reprogramme cells either to expand our understanding of cell biology and its intricacies or to execute desired functions, such as those required in biotechnological and biomedical applications.

CRISPR gene activation (CRISPRa) and inhibition (CRISPRi) have become powerful synthetic tools for modulating endogenous gene expression[1–3]. The coordinated activation and inhibition (CRISPRai) of target genes now allows us to fully explore transcriptional landscapes and modify cellular behaviour[4]. This is especially important in metabolic engineering where metabolic fluxes must be redirected towards a desired product, which is usually achieved by upregulating desired reactions and downregulating competing pathways.

In the simplest form, CRISPRai systems are composed of a single catalytically inactive Cas protein, usually dCas9[5], linked to a transcriptional activator by a direct fusion or via a modified gRNA containing a protein binding aptamer[6–11]. By targeting the Cas protein upstream of the 5′ untranslated region (UTR), expression of that gene can be increased. The Cas protein can then be targeted to the 5′ UTR or coding region to block transcription initiation or elongation, thus reducing expression. However, while this approach results in effective gene activation, gene inhibition is less successful without a transcriptional repression domain, and efficient inhibition may not be possible if optimally positioned PAM sites do not exist[8,12].

A more effective approach for achieving CRISPRai is to employ both transcriptional activation and repression domains. This can be realised using orthogonal Cas proteins, such as dCas9 and dCas12a,

[1]Imperial College Centre for Synthetic Biology, Imperial College London, London SW7 2AZ, UK. [2]Department of Bioengineering, Imperial College London, London SW7 2AZ, UK. [3]These authors contributed equally: William M. Shaw, Lucie Studená. ✉e-mail: r.ledesma-amaro@imperial.ac.uk

with one protein fused to an activator and the other to a repressor[13,14]. These fusion proteins are then targeted to the chosen genes using their cognate gRNAs. Alternatively, a single Cas protein can be used, and instead, modified gRNAs with two orthogonal protein binding aptamers can be used to specifically recruit an activator or repressor to the target genes[3,15–18].

Although these latter approaches lead to more effective CRISPRai, the use of mixed identity gRNAs introduces more complexity, often requiring cumbersome cloning methods. Consequently, multiplexing capacities tend to be low and, for example, in the industrially relevant yeast *Saccharomyces cerevisiae* (*S. cerevisiae*), a maximum of 4 gRNAs have been expressed simultaneously for CRISPRai in an attempt to increase beta-carotene production[18]. This constraint in the number of perturbations that can be made at any one time, limits our cellular engineering ambitions, since, most of the time, desired behaviours are achieved by altering the expression of a large number of targets[4]. Powerful assembly methods have recently been developed to allow the straightforward and rapid assembly of polycistronic arrays for expressing up to 7 mixed identity gRNAs in *E.coli*, 12 Cas9 gRNAs in yeast, 25 Cas12a gRNAs in mammalian cells[19–22]. However, such methods have yet to be applied to the combined expression of activation and repression gRNAs.

An additional desirable feature for any transcriptional regulation method is inducibility. It is known that prolonged transcriptional perturbation of genes can impose a fitness cost, leading to genetic instability and phenotypic loss[23,24]. Furthermore, the regulation of essential genes can be difficult or impossible to modulate continuously or without impacting cell growth[25]. Inducibility of CRISPRai would improve stability and reduce the stringency over target selection when targeting such genes, as cell transformants could be recovered and grown before the system is activated. To date, inducibility of large gRNA arrays for multiplexed CRISPR regulation has only been demonstrated for CRISPRa or CRISPRi, separately[19]. Therefore, to exploit the full potential of CRISPRai, a system that can simultaneously activate and inhibit many genes and only be switched on when required is desirable and should accelerate our ability to study and engineer cells.

Here, we present a method for near leak-free, inducible expression of polycistronic arrays containing up to 24 gRNAs from two orthogonal CRISPR/Cas systems. Using this method, we create a highly tuned and easy-to-use CRISPRai toolkit and a gRNA array assembly strategy for multiplexed transcriptional gene regulation in the industrially relevant yeast, *S. cerevisiae*, which can be controlled in an inducible manner.

Our toolkit combines the orthogonal dCas12a and dCas9 proteins fused to the VP activation and Mxi1 repression domains, respectively, and targets these to specific genomic loci with gRNAs expressed from a single Csy4 processed array that can rapidly be assembled from PCR generated fragments. Inducibility is achieved at the gRNA array, where we have developed a method based on the opposing action of Tet-ON and Tet-OFF systems that represses the entire array in the uninduced state while ensuring efficient array transcription and CRISPRai activity once induced. We have fine-tuned the expression levels of the CRISPR proteins for low metabolic burden, which are delivered on a single genomic integration vector compatible with most common lab strains of *S. cerevisiae* and available with 10 different selectable markers. To demonstrate the system, we targeted 11 genes involved in central metabolism with a single CRISPRai construct, resulting in a 45-fold increase in the production of succinic acid, which could be tightly controlled with the chemical inducer, anhydrotetracycline (aTc).

## Results

### Inducible expression of large polycistronic gRNA arrays
Inducible CRISPR-based systems can be achieved by controlling the expression or state of the Cas protein or the gRNA via an exogenous stimulus, such as a chemical or light[26]. For multiplexed CRISPRai, controlling the activity of the system through the inducible expression

of a polycistronic gRNA array presents itself as promising approach. In this way, CRISPR-based gene activation and inhibition can be regulated through the expression of a single transcript, and Cas protein expression can be tuned to balance CRISPRai performance with fitness. Moreover, induction of the system should not impose a severe burden on the host metabolism, as only transcription of the array (and not translation) is required[27]. Additionally, by modulating the level of gRNA abundance, rather than the active state of the CRISPR components, alternate Cas proteins and their cognate gRNAs can be used where activatable versions are not yet developed, providing a universal approach that should be applicable to most CRISPR-Cas systems.

In order to explore possible strategies for creating inducible polycistronic gRNA arrays, we built on our previous work for assembling and expressing multiple gRNAs from a constitutive, Pol II-driven RNA transcript, which are then processed by the Csy4 endonuclease for multiplexed CRISPRi using dCas9-Mxi1[20]. Based on previous success of expressing individual gRNAs, we decided to develop inducibility using the Tet expression system[11,18,26]. However, in the absence of the inducer aTc, where we desire no repression from CRISPRi, our first two designs, which incorporated a low-leak and then leak-free promoter, reduced respective expression of our fluorescent protein reporters to 10% and 54% compared to a no-gRNA control, therefore showing leakiness in the system (Fig. 1a–d, Design 1 + 2). This led us to the key discovery that gRNA arrays can transcribe without a promoter (Fig. 1c, d, No promoter; Supplementary Fig. 1). Since gRNA arrays that target promoters are themselves made of 20 bp fragments of those promoters, we reasoned that these short sequences are sufficient to clear nucleosomes. This may allow the transcriptional machinery to gain access and initiate transcription from within the array, although further investigation is required to characterise the exact mechanisms by which expression of the gRNA array occurs. We therefore needed a method to repress transcription along the entire length of the array, in a way that would also be scalable to widely varied numbers of gRNAs.

To solve this problem, we redesigned the system to now focus on silencing the array instead of the upstream promoter in the uninduced state using the opposing actions of an orthogonal Tet-ON and Tet-OFF system. (Fig. 1c, Design 3). The Tet-ON system is composed of the reverse TetR protein fused to the Gal4 transcriptional activation domain (rtTA-Gal4)[28]. This protein binds to Tet operator (TetO) sites upstream of the 5′UTR in the presence of inducer to drive expression of the gRNA array. The Tet-OFF system uses a mutated version of the TetR protein (E37A P39K) fused to the chromatin remodelling repression domain Mxi1 (mutTetR-Mxi1), and binds to an orthogonal TetO variant sequence (Tet4C5G, mutTetO)[29]. We specifically target the mutTetR-Mxi1 protein to surround clusters of gRNAs to silence transcription across the entire array in the absence of inducer, without recruiting rtTA-Gal4 to these sites and interfering with array transcription, with the upstream sites targeted to a core promoter adapted from Chen et al.[30].

The inducible gRNA array method removed almost all unwanted CRISPRi repression in the uninduced state, resulting in 96–98% of maximum reporter expression in the absence of aTc, demonstrating efficient silencing of the array from mutTetR-Mxi1 when interspersed between groups of gRNAs (Fig. 1d, Design 3). Strong silencing of the array is achievable with up to 6 gRNAs between mutTetO sites, with a small increase in basal CRISPRi activity above this number (Supplementary Fig. 2a, b). Additionally, no significant difference was seen between the induced state and constitutive array expression, showing release of mutTetR-Mxi1 and the recruitment of rtTA-Gal4 to the promoter after addition with $1\,\mu M$ aTc is highly efficient. Together, this resulted in up to 111-fold change in fluorescent protein expression after induction. Furthermore, the repression of the gRNA array in the uninduced state led to reduced growth defects after transformation compared to constitutive expression of the array, presumably due to the lack of dCas9-Mxi1 targeting native genes in the uninduced state (Supplementary Fig. 2c, d).

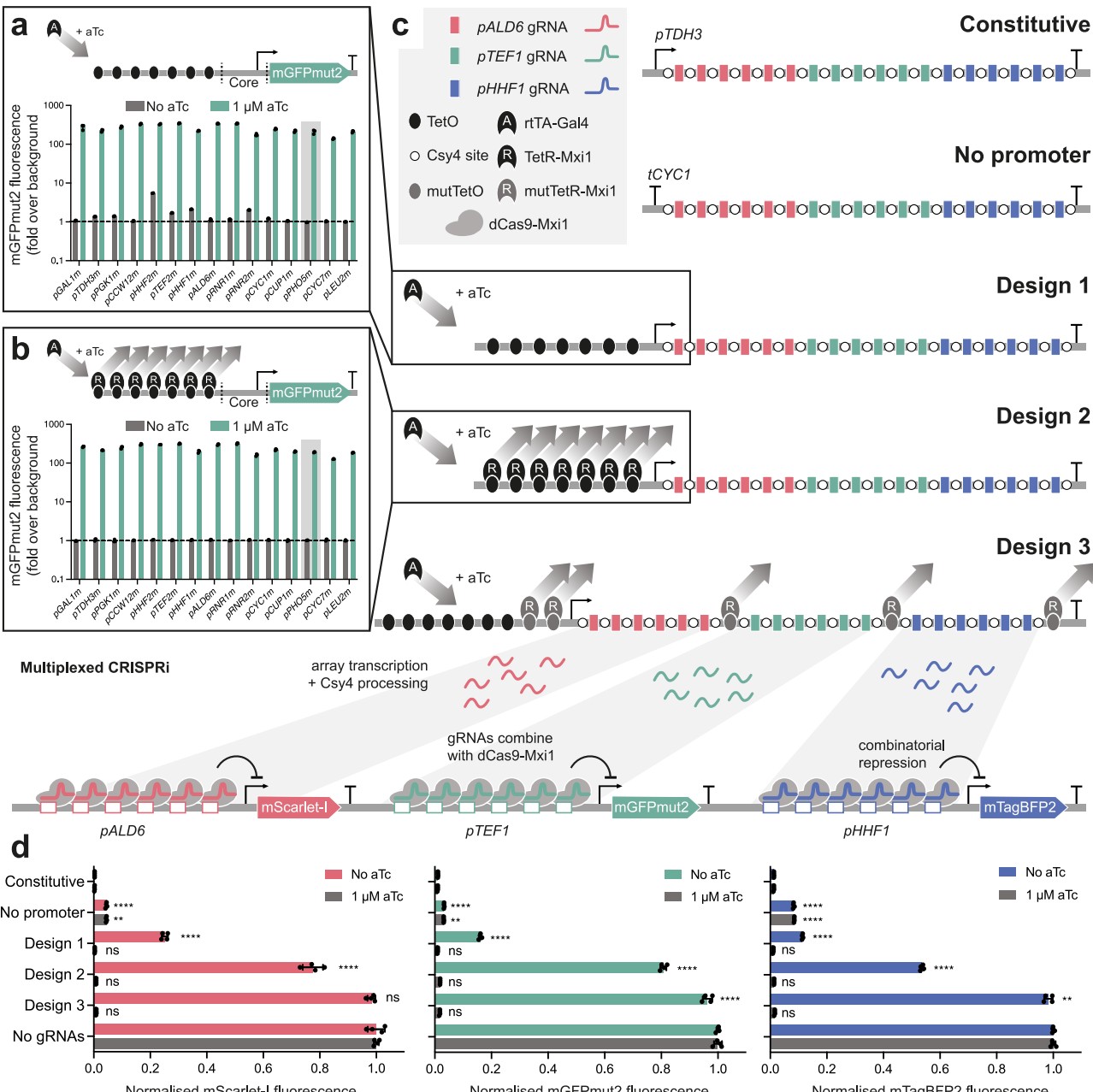

**Fig. 1 | Development of inducible gRNA arrays. a** Development of a low-leak, aTc inducible promoter. rtTA-Gal4 targeting 7x TetO sites upstream of a core promoter library driving the expression of mGFPmut2. Addition of 1 μM aTc recruits rtTA-Gal4 to the promoter, upregulating the expression of mGFPmut2. The *PHO5* minimal core promoter (*pPHO5m*) exhibited the lowest level of basal activity in the absence of inducer, and was used in the initial inducible gRNA array, Design 1. **b** Development of a leak-free, aTc inducible promoter. The aTc-repressible TetR protein fused to the strong transcriptional repressor Mxi1 (TetR-Mxi1) was introduced alongside rtTA-Gal4 to bind TetO sites in the absence of inducer and repress low levels of basal transcription in the off state[41], reducing all promoters to undetectable levels of mGFPmut2 fluorescence in the absence of inducer. The low-leak 7xTetO-*PHO5m* promoter was again chosen to build the second iteration of the

inducible gRNA array, Design 2. **c** Csy4 processed gRNA arrays driven by the various expression systems. Arrays are composed of 18 gRNAs designed to target the constitutive yeast *ALD6*, *TEF1*, and *HHF1* promoters driving the expression of mScarlet-I, mGFPmut2, and mTagBFP2, respectively, for repression by dCas9-Mxi1. **d** Fluorescence measurements of mScarlet-I, mGFPmut2, and mTagBFP2 in the presence and absence of 1 μM aTc across the various gRNA array expression systems, normalised to a no-gRNA and a no-fluorescent protein control. Experimental measurements are fluorescence levels per cell as determined by flow cytometry and shown as the mean ± SD from four biological replicates. Statistically significant differences between Constitutive (1 μM) or No gRNA (No aTc) and all other conditions were tested by 2-way ANOVA, and significance levels are shown as $p < 0.01$ (**), and $p < 0.0001$ (****).

## Design and optimisation of the inducible CRISPRai toolkit

After developing the inducible gRNA array method with CRISPRi (gene repression), we next introduced a CRISPRa (gene activation) protein to complete the inducible CRISPRai toolkit. Building upon the previous work of Lian et al, who demonstrated the use of orthogonal Cas proteins to coordinate the up- and down-regulation of two target genes in

yeast, we introduced the nuclease-deficient Cas12a from *Lachnospiraceae bacterium*, fused to the VP transcriptional activation domain, to play the role of activator (dCas12a-VP)[13]. As CRISPR proteins are known to cause toxicity at high levels[31,32], we decided to explore the effect of protein expression on CRISPRai performance and cell fitness. We combinatorially varied the expression levels of dCas12a-VP,

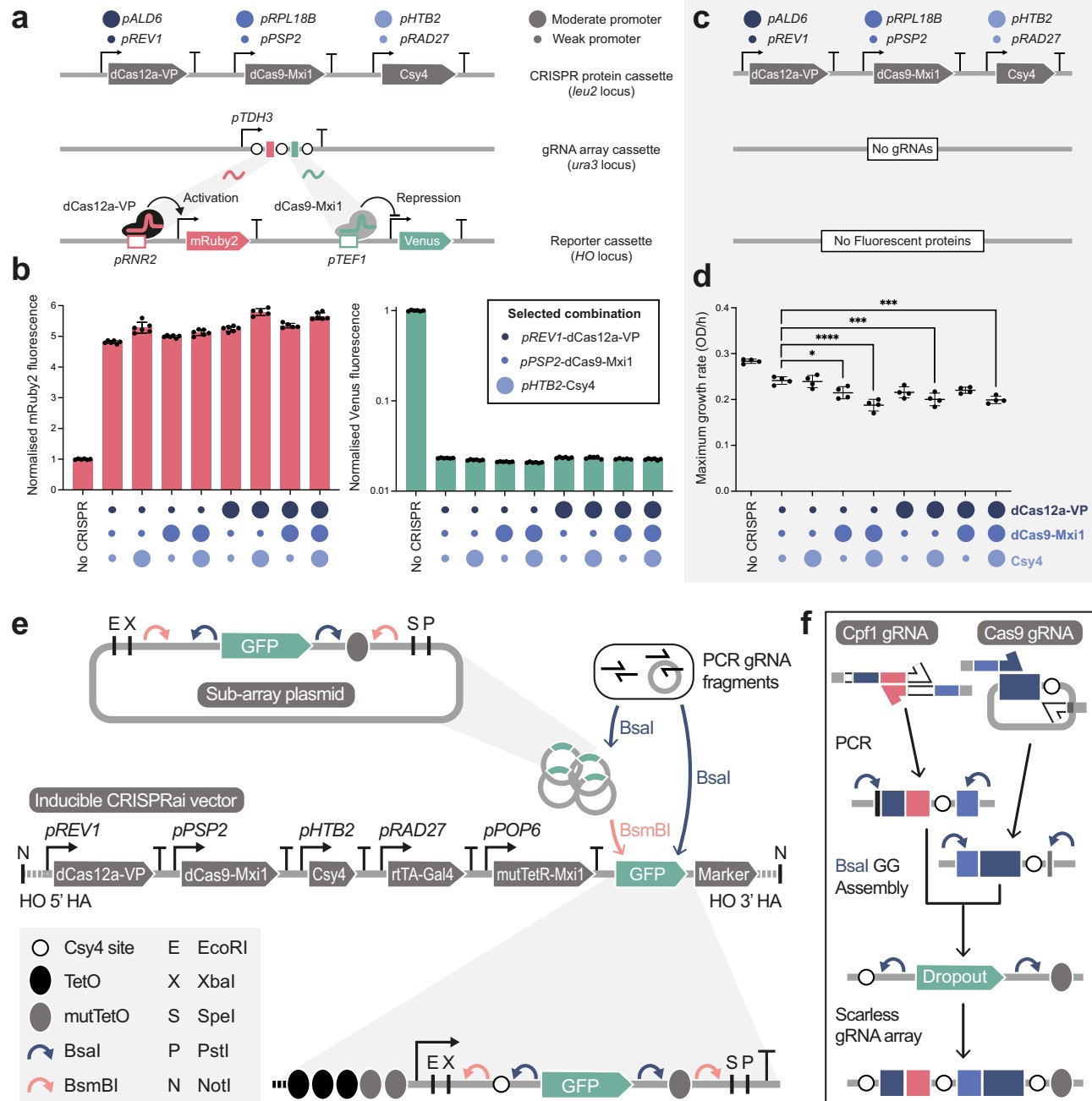

**Fig. 2 | CRISPR protein expression optimisation and inducible CRISPRai toolkit architecture. a + b**, Impact of CRISPRai protein expression on gene activation and repression. **a** Low (small circle) and medium (large circle) strength promoter combinations used to drive the expression of dCas12a-VP (dark blue), dCas9-Mxi1 (blue), and Csy4 (light blue). The constitutive *TDH3* promoter was used drive the transcription of an array containing an activation (dCas12a-VP) and repression (dCas9-Mxi1) gRNA targeting the *RNR2* and *TEF1* promoters driving the expression of mRuby2 and Venus, respectively. **b** Fluorescence measurements of all mid and low strength promoter-CRISPR protein combinations normalised to a control with no CRISPR proteins (No CRISPR). Experimental measurements are mRuby2 and Venus fluorescence levels per cell as determined by flow cytometry and shown as the mean ± SD from six biological replicates. **c + d**, Impact of CRISPRai protein

expression on growth. **c** Low and medium strength promoter combinations used to drive the expression of the CRISPR proteins. **d** Maximum growth rates of all low and medium strength promoter combinations driving the expression of the CRISPR proteins and compared to a control with no CRISPR proteins (No CRISPR). Results are calculated maximum growth rates in YPD medium as determined from growth curves in a plate reader at OD$_{700}$ and are shown as the mean ± SD from four biological replicates. **e** Inducible CRISPRai vector architecture and gRNA array assembly. Inducible CRISPRai vector backbone (KanR-ColE1) not shown. **f** PCR generation of gRNA fragments for scarless BsaI Golden Gate assembly of gRNA arrays. Statistically significant differences between the all low strength promoter combination to all other promoter combinations were tested by one-way ANOVA, and significance levels are shown as $p < 0.05$ (*), $p < 0.001$ (***), $p < 0.0001$ (****).

---

dCas9-Mxi1, and Csy4 using low and medium strength promoters from the Yeast MoClo Toolkit[33] and assessed target gene regulation and cell growth (Fig. 2a–d).

To report on CRISPR gene activation and inhibition, we targeted dCas12a-VP and dCas9-Mxi1 to the *RNR2* and *TEF1* promoters driving

the expression of mRuby2 and Venus using a constitutively expressed gRNA array (Fig. 2a). Varying the expression of the three CRISPR proteins had little effect on fluorescence reporter output (No significant effect on maximum repression and a slight effect on maximum activation, between 10-20 %) (Fig. 2b). We also expressed the three CRISPR

proteins in the absence of gRNAs and fluorescent proteins to determine the effect of protein expression on growth (Fig. 2c). As expected, increasing the strength of the promoters driving the expression of these proteins reduced the maximum growth rate (Fig. 2d). Based on these findings, we chose to build the inducible CRISPRai toolkit with the weak *REV1*, *PSP2*, and medium strength *HTB2* promoters driving the expression of dCas12a-VP, dCas9-Mxi1, and Csy4, respectively, as higher expression did not incur a large performance benefit (no change to maximum repression and > 90% of maximum activation) but did lead to a significant fitness cost. As rtTA-Gal4 and mutTetR-Mxi1 were already under the control of the weak *RAD27* and *POP6* promoters, we kept these fixed.

The inducible CRISPRai toolkit consists of an all-in-one genomic integration vector containing the full set of proteins required for inducible CRISPRai and a gRNA array assembly method (Fig. 2e, f). The inducible CRISPRai vector has been designed to integrate at the *HO* locus, which is conserved between common lab strains of *S. cerevisiae*, and is available with 6 auxotrophic and 4 antibiotic selectable markers (*URA3*, *LEU2*, *HIS3*, *TRP1*, *LYS2*, *MET17*, *KanR*, *NatR*, *HygR*, and *ZeoR*), and so should be appropriate for most applications. gRNA arrays are cloned into the vector using PCR generated fragments that are assembled directly into the vector for up to 6 gRNAs in a single round of Golden Gate assembly (Supplementary Fig. 3), or up to 24 gRNAs via four intermediate sub-array plasmids in two rounds of Golden Gate assembly (Supplementary Fig. 4). gRNAs for gene activation (dCas12a-VP) and repression (dCas9-Mxi1) can be organized in any order, and Csy4 sites are positioned scarlessly either side of each guide to ensure processed RNA structures are equivalent. The limit of 6 gRNAs per vector or sub-array (24 gRNAs when sub-arrays are added together) is recommended to ensure a tight off state by keeping the distribution of mutTetO sites within the limits of mutTetR-Mxi1 silencing, and additionally simplifies validation of array identity by Sanger sequencing.

### Application of the CRISPRai toolkit for metabolic engineering

As we anticipate that metabolic engineering will be a major application of the inducible CRISPRai toolkit in yeast, we next sought to assess how the system would perform over time in batch culture, aiming to achieve stable activation and repression over time. We thus designed an experiment to repress and activate fluorescence reporter expression and measure the output at 24-h intervals after a single induction at 0 h. We assembled a CRISPRai array consisting of 3 activation and 3 repression gRNAs targeting the *RNR2* and *TEF1* promoters driving the expression of mScarlet-I and mTagBFP2, respectively, and transformed this into the dual reporter strain (Fig. 3a). 1 day after induction, mScarlet-I expression increased by 800% and mTagBFP2 expression decreased by 90%. Repression and activation were maintained over at least five days (Fig. 3b and Supplementary Fig. 5). Additionally, the array remained stable in the uninduced state over at least a week of daily cell passaging, thus avoiding possible phenotypic loss before the experiment has begun (Supplementary Fig. 6).

To test whether the system can be practically used for increasing the production of metabolites, we constructed an inducible array of 11 gRNAs targeting strategic nodes in central metabolism for repression and activation, based on past publications on succinic acid overproduction in yeast[34–36] (Fig. 3c). The array contains 9 repression gRNAs targeting *ADH1*, *ADH3*, *FUM1*, *IDP1*, *SDH1*, *SDH3*, *SER3*, *SDH2*, and *SER33*, and 2 activation gRNAs targeting *ADR1* and *ICL1* (Targeted) (Fig. 3d). gRNA targets were designed in Benchling, targeting activation gRNAs between −200 and −350 bp and repression gRNAs between −100 and +150 bp relative to the start codon location of the chosen genes. An additional control array was created using repression and activation gRNAs encoding a random spacer sequence that is not present within the *S. cerevisiae* genome, with this confirmed using the Benchling CRISPR tool off-target score and BLAST (Untargeted).

We transformed the arrays into wildtype BY4741 yeast alongside a no-CRISPR control (WT), with the remaining auxotrophic markers introduced on a single-copy plasmid to create fully complemented strains for growth in minimal media[37]. RT-qPCR confirmed we were indeed regulating the 11 genes in the intended manner, albeit to varying extents (Fig. 3e). In the induced state, a 45-fold increase in succinic acid production was seen in the Targeted strain over the WT strain after 2 days in batch culture (WT (induced) = $9.38 \pm 5.7$ mg/L, Targeted (induced) = $426.9 \pm 13.3$ mg/L), representing a 16-fold change in succinic acid when compared to the uninduced Targeted strain ($26.4 \pm 0.5$ mg/L) (Fig. 3f). Finally, no significant difference in succinic acid titres were measured between all conditions excluding the induced Targeted strain, demonstrating that the increase in succinic acid was exclusively caused by CRISPRai and inducibility is highly controlled, as seen in our previous experiments with the regulation of fluorescent protein expression.

## Discussion

In summary, we have developed a method to enable coordinated activation and repression (CRISPRai) of multiple target genes using long gRNA arrays (multiplexed), whose expression can be controlled with an external molecule (inducible).

In this work, we noted that achieving inducibility for CRISPRai systems with long gRNA arrays was not straightforward, since well-established Tet induction systems used to express single gRNAs[11,18,26] fail in long arrays. We found that long gRNA arrays targeting promoter regions (usually the case in CRISPRai studies), can express themselves even with no promoter. Therefore, we created a method that silences the gRNA array when not in use, which led to fewer issues when transforming CRISPRai constructs and better growth and improved genetic stability in the absence of the inducer. These are particularly beneficial properties when targeting genes that play a role in growth or an essential cellular process and should simplify the study and control of gene interactions and networks.

We incorporated this method into a highly tuned and easy-to-use CRISPRai toolkit that can deliver up to 24 gRNAs on a single array for inducible up- and down-regulation of genes in the industrially relevant yeast, *S. cerevisiae*, representing the first system to incorporate simultaneous activation and repression, large multiplexing capacity, and inducibility. Tuning the expression of the CRISPR proteins allowed us to balance performance of the system with metabolic burden. However, a penalty for expressing Cas9-Mxi1, dCas12a-VP, and Csy4 still remains. Future work to determine alternative Cas proteins which impose a reduced fitness cost could be used to further improve the toolkit.

As one of the main chassis organisms for metabolic engineering, tools that can accelerate strain creation or prototype extensive genetic modifications in yeast are greatly needed. Indeed, several CRISPRai toolkits have been established for metabolic engineering in *S. cerevisiae*[8,9,13,18]. However, these toolkits have been unable to target more than four genes for up- or down-regulation, and seeing only modest increases in target product yields, no greater than threefold[8,13,18]. To demonstrate the benefits that can be realised by extending multiplexing capacity for metabolic engineering, we used the CRISPRai toolkit to increase the production of succinic acid.

By targeting 11 genes from central metabolism for activation or repression, we were able to increase succinic acid titres by 45-fold over natural titres. This increase in production was tightly controlled using 1 µM aTc, allowing us to grow strains and then switch to a mode of production at a defined time point. Notably, the entire system was delivered on a single construct, allowing us to generate the new strain in a single transformation, and integrates at a common locus in the *S. cerevisiae* genome with a choice of 10 selectable markers, making it applicable to most strains and applications.

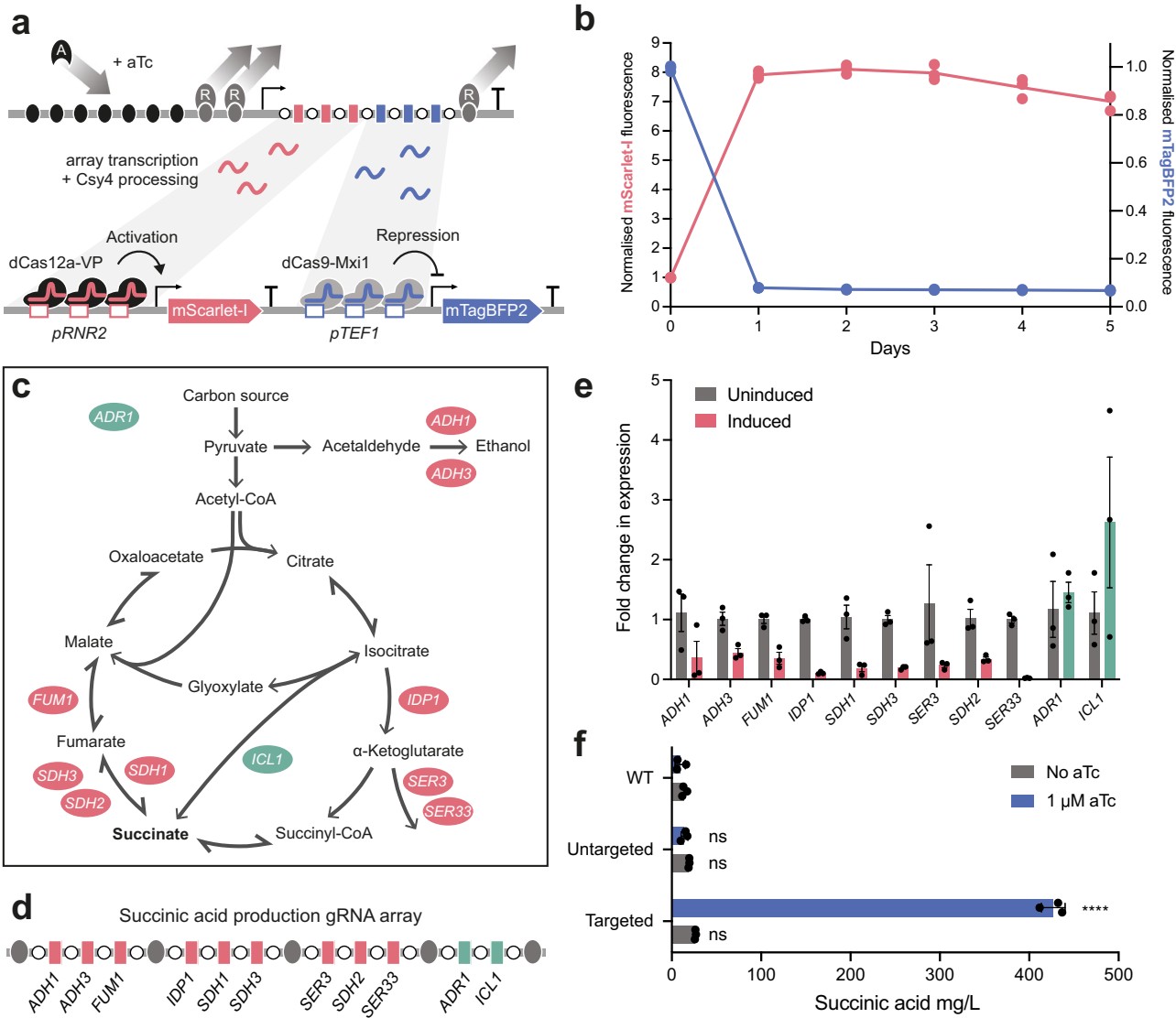

**Fig. 3 | Inducible CRISPRai for metabolic engineering. a** Inducible CRISPRai gRNA array containing three activation gRNAs targeting the *RNR2* promoter (red) followed by three repression gRNAs targeting the *TEF1* promoter (blue). Expression and Csy4 processing of the gRNA array results in the upregulation of mScarlet-I, through the recruitment of dCas12a-VP to the *RNR2* promoter, and down regulation of mTagBFP2 through recruitment of dCas9-Mxi1 to the *TEF1* promoter. **b** Time course of mScarlet-I and mTagBFP2 fluorescence after 1 μM aTc induction at 0 h. Experimental measurements are mScarlet-I and mTagBFP2 fluorescence levels per cell as determined by flow cytometry and shown as individual values from three biological replicates. **c** Overview of succinic acid production and CRISPRai targets. Respective CRISPRa and CRISPRi targets shown in green and red. **d** gRNA array targeting the 11 genes for overproduction of succinic acid. **e** Relative expression of the 11 genes targeted by CRISPRai in the Targeted strain in the presence and absence of inducer. Experimental measurements are relative gene expression levels compared to WT yeast as determined by RT-qPCR and shown as the mean ± SE from three biological replicates. **f** Quantification of succinic acid from WT, Untargeted, and Targeted strains in the presence and absence of inducer. Experimental measurements are succinic acid concentrations as determined by LC-MS and shown as the mean ± SD from three biological replicates. Statistically significant differences between WT (No aTc) or WT (1 μM) and all other conditions were tested by 2-way ANOVA, and significance levels are shown as $p < 0.0001$ (****).

Altogether, this toolkit opens the door to more extensive, flexible, and faster strain construction for accelerating metabolic engineering in yeast. Finally, the underlying principles of inducibility presented here should be applicable to other species across biology and scalable to other CRISPR/Cas systems for increasing the number of orthogonal genetic programs that can be run simultaneously.

## Methods
### Inducible CRISPRai toolkit overview
The inducible CRISPRai toolkit consists of an all-in-one genomic integration vector containing the full set of proteins required for inducible CRISPRai and a GFP dropout in place of the gRNA array (Fig. 2e). gRNA arrays are cloned into the vector using PCR generated fragments that

are assembled directly into the vector for up to 6 gRNAs in a single round of Golden gate assembly (Supplementary Fig. 3), or up to 24 gRNAs via four intermediate sub-array plasmids in two rounds of golden gate assembly (Supplementary Fig. 4). mutTetO sites are included within the inducible CRISPRai vector and sub-array plasmids so that they are distributed throughout the array, and spacers are included in instances where not all 4 sub-array vectors are required (biologically neutral DNA designed by R2oDNA designer[38]). The limit of 6 gRNAs per CRISPRai vector or sub-array plasmid (24 gRNAs when sub-arrays are added together) is recommended to ensure a tight off state by keeping the distribution of mutTetO sites within the limits of mutTetR silencing. This also simplifies validation of array identity by Sanger sequencing. The inducible CRISPRai vector has been designed

to integrate at the *HO* locus, which is conserved between common lab strains of *S. cerevisiae* and is available with 6 auxotrophic and 4 antibiotic selectable markers (*URA3*, *LEU2*, *HIS3*, *TRP1*, *LYS2*, *MET17*, KanR, NatR, HygR, and ZeoR), and so should be appropriate for most strains and applications. For a full list of plasmids in the inducible CRISPRai toolkit, see Supplementary Table 1.

### gRNA target design

All gRNAs were designed in Benchling, using the CRISPR Design Tool. For gene activation gRNAs (dCas12a-VP), targets were chosen between −200 and −350 bp relative to the start codon location of the chosen genes. For repression gRNAs (dCas9-Mxi1), targets were chosen between −100 to +150 bp relative to the start codon location of the chosen genes. All gRNAs used in this study are listed in Supplementary Table 2. 20 bp target sequences cannot contain internal BsaI, BsmBI, or NotI restriction sites, required for downstream cloning and transformation purposes. Additionally avoiding EcoRI, XbaI, SpeI, and PstI is useful for downstream cloning of fully assembled arrays using the BioBrick assembly method and digest verification (see additional cloning features below), although not necessary.

### gRNA array design

To generate the gRNA fragments for array assembly, primer pairs were designed to amplify without a template for activation gRNAs and with a template (pWS3799 – Cas9 gRNA-Csy4 template) for repression gRNAs (Supplementary Fig. 3a). Each dsDNA gRNA fragment includes the Cas protein specific gRNA scaffold, 20 bp target sequence, and a Csy4 site at the 3′ end. BsaI-generated overhangs within the CRISPRai vector and sub-array plasmids occur within the Csy4 site at the start and mutTetO site at the end of the array, and by designing the BsaI overhangs to occur within adjacent gRNA fragments, gRNA arrays can be made scarlessly. This creates an array of gRNAs each flanked precisely by Csy4 sites (Supplementary Fig. 3c). There are no constraints on the organization of gRNAs within the array, and activation and repression gRNAs can be designed in any order. Note: Resulting arrays are highly repetitive, particularly around the Cas9 gRNA handle (depending on the number of gRNAs in the final array). Although this was not seen during batch culture, arrays can recombine over multiple cell passages while induced (Supplementary Fig. 6).

### Activation (dCas12a-VP) gRNA fragment PCR

Activation gRNA PCRs were set up in 20 μL volume reactions, as follows: 4 μL of 5x Q5 Reaction Buffer (NEB), 0.4 μL of 10 mM dNTPs (NEB), 1 μL of each primer (100 μM), 0.2 μL of Q5 High-Fidelity DNA Polymerase (NEB), and 13.4 μL ddH$_2$O. Activation gRNAs were created in 5 cycles of a non-amplifying extension PCR reaction, as follows: 30 s at 98 °C, (10 s at 98 °C, 20 s at 61 °C, 30 s at 72 °C) x 5 cycles, 30 s at 98 °C, hold at 10 °C.

### Repression (dCas9-Mxi1) gRNA fragment PCR

Repression gRNA PCRs were set up in 20 μL volume reactions, as follows: 4 μL of 5x Q5 Reaction Buffer (NEB), 0.4 μL of 10 mM dNTPs (NEB), 1 μL of each primer (100 μM), 1 μL of pWS3799 plasmid (~1 ng/μL), 0.2 μL of Q5 High-Fidelity DNA Polymerase (NEB), and 12.4 μL ddH$_2$O. Repression gRNAs were generated in a standard, 30-cycle amplifying PCR reaction, as follows: 30 s at 98 °C, (10 s at 98 °C, 20 s at 57 °C, 30 s at 72 °C) x 30 cycles, 30 s at 98 °C, hold at 10 °C. DpnI digestion of the template DNA is not required following the PCR reaction.

### gRNA fragment purification

To purify gRNA fragments after PCR, 4 μL of 6x loading dye (NEB) was added to the completed reaction and run on a 2% agarose gel until total separation of DNA bands. After gel electrophoresis, gel bands were excised and DNA was extracted using Zymoclean Gel DNA Recovery kit (Zymo Research), following manufacturer instructions. As gRNA

fragments are small (~100 bp for activation gRNAs and ~150 bp for repression gRNAs), it is important to excise a clean band from the gel, avoiding residual primer sequences which will run close to the desired band. Once purified, gRNA fragment DNA concentration was measured (NanoDrop One) and samples were diluted to 100 fmol/μL.

### gRNA fragment assembly into the CRISPRai vector/sub-array plasmid

gRNA fragments were assembled into the CRISPRai vector and sub-arrays plasmids in a 20 μL BsaI Golden Gate reaction, using the following setup: 1 μL of CRISPRai vector/sub-array plasmid (50 fmol/μL), 1 μL of each gRNA fragment (100 fmol/μL), 2 μL of T4 DNA ligase buffer (NEB), 1 μL of T4 DNA ligase (NEB), 1 μL of BsaI-HF v2 (NEB), and up to 20 μL with ddH$_2$O. Reaction mixtures were then incubated in a thermocycler using the following program: (37 °C for 5 min, 16 °C for 5 min) x 30 cycles, followed by a final digestion step of 55 °C for 10 min, and then heat inactivation at 80 °C for 10 min, hold at 10 °C. Reactions were then transformed into *E.coli*. Plasmid DNA from GFP negative colonies was isolated by miniprep, screened for the correct array length by colony PCR and then sent for Sanger sequencing to confirm identity.

### Sub-array assembly into CRISPRai Vector

Sub-arrays and spacers were assembled into the CRISPRai vectors in a 10 μL BsmBI Golden Gate reaction, using the following setup: 0.5 μL of CRISPRai vector/sub-array plasmid (50 fmol/μL), 1 μL of each sub-array/spacer (50 fmol/μL), 1 μL of T4 DNA ligase buffer (NEB), 0.5 μL of T4 DNA ligase (NEB), 0.5 μL of BsmBI v2 (NEB), and 3.5 μL of ddH$_2$O. Reaction mixtures were then incubated in a thermocycler using the following program: (42 °C for 2 min, 16 °C for 5 min) x 25 cycles, followed by a final digestion step of 55 °C for 10 min, and then heat inactivation at 80 °C for 10 min, hold at 10 °C. Reactions were then transformed into *E.coli*. Plasmid DNA from GFP negative colonies was isolated by miniprep and screened for the correct array length by colony PCR or restriction digesting using EcoRI/XbaI and SpeI/PstI.

### Additional CRISPRai toolkit cloning features

To increase flexibility of the toolkit once gRNA arrays have been assembled into the CRISPRai vector, a BioBrick cloning prefix (excluding NotI) was included between the promoter and the start of the gRNA array, and a BioBrick cloning suffix (excluding NotI) was included between the end of the gRNA array and terminator. This allows the user to excise and ligate validated gRNA arrays into different CRISPRai vectors to change the yeast selection marker without recreating the array from scratch. Additionally, gRNA arrays can be concatenated by BioBrick assembly to create new combinations of previously assembled arrays without redesigning or assembling new gRNA fragments.

### Strains and cultivation conditions

*E. coli* DH5α was used for propagating all plasmids and grown at 37 °C in Luria Broth (LB) medium containing the appropriate antibiotics for plasmid selection (ampicillin 100 μg/mL, chloramphenicol 34 μg/mL, or kanamycin 50 μg/mL). *S. cerevisiae* strain BY4741[39] (*MATa his3Δ1 leu2Δ0 met15Δ0 ura3Δ0*) was used for all yeast experiments. For succinic acid experiments, fully complemented yeast strains were created by restoring the missing auxotrophic markers on a single-copy plasmid[37]. Yeast extract peptone dextrose (YPD) was used for culturing cells in preparation for transformation: 1% (w/v) Bacto Yeast Extract (Merck), 2% (w/v) Bacto Peptone (Merck), 2% glucose (VWR). Fluorescent reporter assay experiments were performed in synthetic complete (SC) medium: 2% (w/v) glucose (VWR), 0.67% (w/v) Yeast Nitrogen Base without amino acids (Sigma), 0.14% (w/v) Yeast Synthetic Drop-out Medium Supplements without histidine, leucine, tryptophan, and uracil (Sigma), 20 mg/L uracil (Sigma), 100 mg/L

leucine (Sigma), 20 mg/L histidine (Sigma), and 20 mg/mL tryptophan (Sigma). Succinic acid production experiments were performed in synthetic minimal (SD) medium: 2% (w/v) glucose (VWR), and 0.67% (w/v) Yeast Nitrogen Base without amino acids (Sigma).

## Yeast Transformations

For transformation into yeast, 200 ng of the final CRISPRai plasmid was digested at 37 °C for 1 h with NotI in the following setup: 200 ng CRISPRai, 1 μL CutSmart Buffer (NEB), 0.2 μL NotI-HF (NEB), up to 10 μL $H_2O$. Digestions were heat inactivated at 65 °C for 20 min before transformation. Chemically competent yeast cells were transformed using the lithium acetate protocol from Gietz and Schiestl[40], as follows: Yeast colonies were grown to saturation overnight in YPD. The following morning the cells were diluted 1:100 in 15 mL of fresh YPD in a 50 mL conical tube and grown for 4-6 h to $OD_{600}$ 0.8-1.0. Cells were pelleted and washed once with 10 mL 0.1 M lithium acetate (LiOAc) (Sigma). Cells were then resuspended in 0.1 M LiOAc to a total volume of 100 μL/transformation. 100 μL of cell suspension was then distributed into 1.5 mL reaction tubes and pelleted. Cells were resuspended in 64 μL of DNA/salmon sperm DNA mixture (10 μL of boiled salmon sperm DNA (Invitrogen) + DNA + $ddH_2O$), and then mixed with 294 μL of PEG/LiOAc mixture (260 μL 50% (w/v) PEG-3350 (Sigma) + 36 μL 1 M LiOAc). The yeast transformation mixture was then heat-shocked at 42 °C for 40 mins, pelleted, resuspended in 200 μL sterile $H_2O$ and plated onto the appropriate selection medium.

## Inducible CRISPRai toolkit construction

All constructs were created within the Yeast MoClo Toolkit[33] framework and assembled by Golden Gate assembly. Novel parts were synthesized (IDT) or assembled from PCR generated fragments designed using the Benchling Golden Gate tool. All DNA for Golden Gate reactions was set to equimolar concentrations of 50 fmol/μL prior to experiments. Golden Gate reactions were prepared as follows: 0.25 μL of backbone plasmid, 0.5 μL of each DNA fragment or plasmid, 1 μL T4 DNA ligase buffer (NEB), 0.5 μL T4 DNA Ligase (NEB), 0.5 μL restriction enzyme (BsaI-HF v2/BsmBI v2) (NEB), and $H_2O$ to bring the final volume to 10 μL. Reaction mixtures were then incubated in a thermocycler using the following program: (42 °C for 2 min, 16 °C for 5 min) x 25 cycles, followed by a final digestion step of 55 °C for 10 min, and then heat inactivation at 80 °C for 10 min, hold at 10 °C.

## Fluorescent reporter assay

All reporter strains were picked into 500 μL of synthetic complete (SC) medium and grown in 2.2 mL 96 deep-well plates at 30 °C in an Infors HT Multitron, shaking at 700 rpm overnight. The next day, saturated strains were diluted 1:100 into fresh media, with and without 1 μM aTc (Alfa Aesar, J66688-MB). For single-point measurements, cultures were incubated for 16 h and cell fluorescence was measured by an Attune NxT Flow Cytometer (Thermo Scientific). Batch culture and daily cell passaging assay experiments as described in the text. Attune NxT Flow Cytometer settings: FSC 300 V, SSC 350 V, BL1 500 V, VL2 450 V, YL2 450 V. Fluorescence data was collected from at least 10,000 cells for each experiment and analysed using FlowJo software. Note: 1 μM (463 ng/μL) aTc was used, rather than the standard 100 ng/μL, to ensure ligand saturation and full release of the mutTetR-Mxi1 protein from the array. 1000 x stock solution of aTc (1 mM) was in 100% DMSO. Final concentration of DMSO was present in all conditions.

## Growth curves

Single colonies of yeast strains were grown to saturation overnight in 2 mL YPD. The next day, the yeast cultures were back diluted to an $OD_{700}$ of 0.175, and 99 μL was transferred to a 96-well clear, flat-bottom microplate (Corning). $OD_{700}$ was then measured over 24 h by a SpectraMax plate reader (Molecular Devices) taking measurements every 10 min with shaking at 30 °C in between readings. Maximum growth rate was then calculated according to the equation $(\ln(OD600(t+1)/OD600(t)))$, where $t$ is time in hours.

## RT-qPCR

All quantitative PCR (qPCR) strains were picked into 5 mL of synthetic complete (SC) medium and grown at 30 °C, 250 rpm overnight. The next day, optical density was measured in a spectrophotometer (WPA Biowave II) and cultures were diluted to $OD_{600} = 0.05$ in 5 mL SC media, with and without 1 μM aTc (Alfa Aesar, J66688-MB). For RNA purification, RNA was isolated from yeast culture grown to an $OD_{600}$ of $1 \pm 0.1$ using a RiboPure Yeast kit (Invitrogen). RNA was quantified by nanodrop spectrophotometer (Thermo Fisher), and cDNA was generated from each RNA prep using a High-Capacity cDNA Reverse Transcription Kit (Applied Biosystems). Each qPCR reaction contained 10 ng of cDNA. qPCR results were normalized to the housekeeping gene *UBC6*. All qPCR primers were designed manually using Benchling. All quantitative PCR (qPCR) reactions were performed in an StepOnePlus™ Real-Time PCR System (Applied Biosystems) using SYBR Green JumpStart Taq ReadyMix (Sigma-Aldrich).

## Succinic acid production, sampling, and measurement

All succinic acid production strains were picked into 6 mL of synthetic minimal (SD) medium and grown at 30 °C, 250 rpm overnight. The next day, optical density was measured in a spectrophotometer (WPA Biowave II) and cultures were diluted to $OD_{600} = 0.05$ in 1 mL SD media, with and without 1 μM aTc (Alfa Aesar, J66688-MB). Cultures were grown in 48-deep-well-plates (Agilent, 201238-100) at 30 °C in an Infors HT Multitron, shaking at 700 rpm. After 2 days, plates were spun down at $4000 \times g$, 4 °C for 10 min. Then, 300 μL of the supernatant was sampled for each well. The same day, supernatant samples were measured directly by LC-MS alongside a succinic acid standard, as follows: succinic acid was detected and measured by UPLC-MS, using an Agilent 1290 Affinity chromatograph linked to an Agilent 6550 Q-ToF mass spectrometer. Separation was achieved using an Agilent Zorbax Eclipse Plus C18 column (2.1 × 50 mm, 1.8 μm) and an acetonitrile gradient of 0% for 2 min then an increase to 98% over 0.5 min at a flow rate of 0.3 mL/min. Mass spectral data was acquired in negative ion mode from m/z 90 to 1000 at the rate of 3 spectra per second throughout the separation. In total, 0.2 μL was injected from both sample wells and standard solutions. Succinic acid concentrations were calculated from a succinic acid standard curve in Microsoft Excel.

## Statistics and reproducibility

Unless otherwise stated, all data was analysed in Prism (GraphPad). Error bars represent the standard deviation/error as noted in the figure legend and ANOVA was used for statistical analyses with Prism (GraphPad) where significance is noted ($p < 0.05$). The respective number of replicates are given in the figure legend and all replicates are included in the manuscript.

## Reporting summary

Further information on research design is available in the Nature Research Reporting Summary linked to this article.

## Data availability

CRISPRai toolkit plasmids and their nucleotide sequences are available through Addgene for distribution (Addgene ID listed in Supplementary Table 1). gRNA spacer sequences for used in all experiments are available in Supplementary Table 2. Additional plasmids and strains used in this study are available from the corresponding author upon reasonable request, who will respond within a couple of weeks of the request and explain the next steps required to sign an MTA. Individual data points for all graphs are provided as source data with this paper. Source data are provided with this paper.

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

## Acknowledgements
R.L-A. received funding from BBSRC (BB/R01602X/1), (19-ERACoBio-Tech- 33 SyCoLim BB/T011408/1 and BBSRC BB/T013176/1) British Council 527429894, Newton Advanced Fellowship (NAF\R1\201187), Yeast4Bio Cost Action 18229, European Research Council (ERC) (DEUSBIO – 949080) and the Bio-based Industries Joint (PERFECOAT-101022370) under the European Union's Horizon 2020 research and innovation programme. The authors wish to thank David Bell and Syn-BiCITE for providing support with the LC-MS.

## Author contributions
W.M.S., L.S., K.R., N.S.M., and R.L.-A. designed the experiments. W.M.S., L.S., K.R., P.H., N.S.M., and A.E.G., performed the experiments. W.M.S., L.S., and K.R., performed the data analysis. W.M.S., L.S., K.R., N.S.M., A.E.G., T.E., and R.L.-A interpreted the results. W.M.S. and L.S. wrote the manuscript. All authors reviewed and approved the final manuscript.

## Competing interests
The technology described in this work is part of a patent application by Imperial College London, whose authors are R.L-A., T.E., W.M.S., and L.S. Patent application number GB2201404.7 (status at the time of publication: application filed).
