## [Peer Review File · Nature Communications]

Reviewers' Comments:

Reviewer #1:

Remarks to the Author:

In this work, Shaw and coworkers combine inducible large gRNA arrays with CRISPR nucleases for activation and inhibition (CRISPRai) in yeast. The authors obtained substantial activity even when replacing the inducible promoter with a terminator, requiring a distinct approach to minimize uninduced activity. The resulting design, which included tet activation and repression within the promoter and the gRNA array, offered a large fold-change, with minimal leakiness. The authors then applied this design to modulate the expression of genes involved in succinate metabolism, leading to a 24-fold improvement in succinate production compared to a WT strain.

CRISPRai as well as gRNAs arrays have been used to modulate the expression of multiple genes as noted by the authors. However, the strength of the work is tackling two key issues (creating large gRNA arrays and ensuring tight inducible control) that result in impressive improvements in the number of encoded gRNAs through a manageable cloning approach, the tightness of inducible control, and the overall stability of the CRISPRai system over time. The inclusion of an application boosting succinate production in yeast nicely wrapped up the work. Given these improvements, I see this work as a useful advance, even if the individual components were previously established. The work does have a few conceptual and technical holes that need to be filled to fully support some conclusions and the impact of the work.

Major comments:

1. There is a large jump from Designs 1 and 2 to Design 3, leaving it unclear whether the use of the operators in the promoter or in the arrays account for the difference. It's possible that only one of the two locations was sufficient for tight inducible control.
2. In multiple places (e.g. lines 107, 213), the authors state the discovery that gRNA arrays can be transcribed without a promoter and propose that the promoter-matching guides are driving transcription. I found this claim dubious and in need of supporting data to stand. For one, the introduced terminator may be leaky (especially when combined with upstream transcription in the plasmid) or could include a cryptic promoter, both of which would mean the selected spacers have nothing to do with leakage. This could be explored using encoded reporters or RNA quantification (e.g. RT-qPCR, RNA-seq) of the transcribed array.
3. One of the central arguments is that using an inducible gRNA array helped stabilize the CRISPRai construct and minimize any growth defects. There are a few aspects that need to be addressed and improved. First, the growth measurements in Figure 2D need statistical analyses and would be stronger if the experiment was conducted as a competition (e.g. mixing barcoded strains with the different constructs). Second, the demonstration of succinate production would be more compelling if the authors could show that one of the early designs (or constitutive gRNA array expression) is less stable or exhibits slower growth than the optimal design—specifically under uninduced conditions.

Other comments:

4. Line 31: Could you use "coordinated" rather than "simultaneous"? "simultaneous" can be understood to mean that one gene is being both activated and repressed.
5. Line 53: constraint rather than constrain
6. Line 82: specify that you're using strains of yeast.
7. Introduction: I recommend citing PMID = 31270316 as an original example of combining Cas9 and Cas12a gRNAs in the same array.
8. Line 91 – 93: More accurate phrasing would be helpful here, as the entire system isn't regulated (CRISPRai is but not the Cas proteins), and the levels of the Cas proteins (versus the target

genes) can be tuned.

9. Line 106: in comparison to a no-gRNA control.

10. Line 118: I recommend commenting on the mechanism of repression so it's clear how Mxi1 would impact transcription not only downstream but also upstream of the operator site.

11. Line 200 – 206: the uninduced condition is brought up two separate places in the paragraph. Can the logic be slightly restructured so the uninduced condition is addressed only once?

12. Figure 2F: can you introduce gray ends on the primers to exactly match the resulting dsDNA fragment?

13. Figure 3B: I recommend adding a colored dot with each vertical axis so it's more obvious which line goes with which axis.

14. Suppl. Figure 3: The bottom segment of each plot appears to be missing a label.

Reviewer #2:

Remarks to the Author:

The manuscript presents a CRISPR/Cas-based approach to bi-directional transcriptional regulation in yeast with inducibility of large gRNA arrays. The presented work is scientifically sound and demonstrates 8- and 9-fold bi-directional transcriptional regulation of at least two distinct genes simultaneously. The system leakiness was reduced with design 3 while maintaining high inducibility. Tuning of Cas protein expression was investigated for reducing metabolic burden, and a clear design- and cloning strategy for constructing large gRNA arrays is included. The main novelty that the authors claim is first use of simultaneous transcriptional activation and repression in combination with large gRNA arrays and inducibility.

Simultaneous transcriptional activation and repression is established literature, and so is multi-gene regulation with large gRNA arrays and inducibility of such. While I do appreciate the efforts to reduce system leakiness, metabolic burden to the cell, and making the system easy to clone and use in yeast, I find that the novelty does not merit publication in the current journal.

Major recommendations:

1. I suggest that you use ANOVA for statistical analyses with Prism (GraphPad) where relevant.

2. With regards to Fig. 3d+e you introduce 11 gRNAs targeting central metabolism for repression and activation to achieve 45-fold improvement of succinic acid titers relative to wild-type. This strategy shows that in situ design and introduction of multiple gRNAs is applicable to improving cell factories, without testing their individual activity first. However, the output (succinic acid) does not give information on how many promoters were regulated during system induction and to what extent.

This is a problem mainly because the manuscript, in my opinion, demonstrates bi-directional transcriptional regulation with certainty of no more than two genes simultaneously at this point.

Further, the fluorescence analysis on expressing multiple gRNAs in Supplementary Fig. 1b is unable to give a complete insight into expression and activity of individual gRNAs when they are expressed in combination. In fact, repression of mTagBFP2 appears to be slightly relieved with 7 or 8 gRNAs targeting the HHF1 promoter being co-expressed. I miss clear evidence that all gRNAs in a 24-gRNA array are indeed functionally expressed, and to what extent. For example, Zhang et al. 2019 (<https://doi.org/10.1038/s41467-019-09005-3>) noticed that expressing >5 gRNAs in array in *S. cerevisiae* resulted in decreased frequency of gene disruption, and they speculate that low gRNA expression is the underlying cause (granted, a different gRNA expression strategy was applied).

At the very least, I suggest making a comparative analysis clearly showing the transcript level for all 11 target genes in targeted and untargeted strains used for quantifying succinic acid titers.

Alternatively, and even better for claiming full gRNA array activity, prior to showing Fig. 3d+e, you may consider targeting 24 different promoters simultaneously for activation and repression and analyze transcript levels of these genes relative to an untargeted control.

As you perform your analysis, pay attention to both activity and position for each gRNA in the array. Is there a trend here that may influence how you and others will design large gRNA arrays in future?

3. Can you give some consideration in your discussion to the persistent leakiness of the system (comparing titers presented in line 204) in relation to metabolic burden and possibly how to further avoid it?

Minor recommendations:

1. Cite Qi et al. 2013 for coining the term CRISPRi and for dCas9. doi: 10.1016/j.cell.2013.02.022.
2. Line 35: Insert citation(s) (e.g., Zalatan et al. 2015 doi: 10.1016/j.cell.2014.11.052)
3. Line 73: Introduce abbreviation for *Saccharomyces cerevisiae* (*S. cerevisiae*) and use throughout.
4. Fig. 2b: I would like the promoter-codes nuances to be clearer/more distinct.
5. Fig. 2d: Do growth rates differ significantly?
6. Line 200-201: "Fig. 3f" should be corrected to Fig. 3e.
7. Line 203-204: "No major differences". Replace with statistic measures.
8. Line 342: Use subscript for "600" in "OD600".
9. Line 325: Insert citation for BY4741.

Sincerely,

Emil D. Jensen

Response to reviewer comments for Shaw and Studená *et al.*

We wish to thank both reviewers for providing valuable time towards helping improve our manuscript and assess its strengths and weaknesses. This has now led to a significantly improved manuscript that benefits from clearer statements and revised experimental results and additional figures. A summary of the main changes:

- New experiment to bolster the evidence that gRNA arrays can express without a promoter (Supplementary Fig. 1)
- Additional work to show the benefits of inducibility of CRISPRai over constitutive activity (Supplementary Fig. 2d)
- Expression profiling of the 11 genes in targeted to increase the production of succinic acid (Fig. 3e)
- Statistical analysis of all relevant data to assess the performance of the system and help explain the motivations behind key decisions made throughout the work
- Changes throughout the introduction, results, and discussion to include this additional work, improve the clarity of the study, and incorporate other reviewer feedback

Below we address issues raised in the reviews from each referee point-by-point.

Reviewer #1 (Remarks to the Author):

In this work, Shaw and coworkers combine inducible large gRNA arrays with CRISPR nucleases for activation and inhibition (CRISPRai) in yeast. The authors obtained substantial activity even when replacing the inducible promoter with a terminator, requiring a distinct approach to minimize uninduced activity. The resulting design, which included tet activation and repression within the promoter and the gRNA array, offered a large fold-change, with minimal leakiness. The authors then applied this design to modulate the expression of genes involved in succinate metabolism, leading to a 24-fold improvement in succinate production compared to a WT strain.

CRISPRai as well as gRNAs arrays have been used to modulate the expression of multiple genes as noted by the authors. However, the strength of the work is tackling two key issues (creating large gRNA arrays and ensuring tight inducible control) that result in impressive improvements in the number of encoded gRNAs through a manageable cloning approach, the tightness of inducible control, and the overall stability of the CRISPRai system over time. The inclusion of an application boosting succinate production in yeast nicely wrapped up the work. Given these improvements, I see this work as a useful advance, even if the individual components were previously established. The work does have a few conceptual and technical holes that need to be filled to fully support some conclusions and the impact of the work.

Major comments:

1. There is a large jump from Designs 1 and 2 to Design 3, leaving it unclear whether the use of the operators in the promoter or in the arrays account for the difference. It's possible that only one of the two locations was sufficient for tight inducible control.

Thank you for highlighting what is seemingly a large conceptual leap in our designs. There are indeed two big changes between design 2 and 3; i) we removed repression of the UAS, as seen in design 2, which is now only targeted exclusively by rtTA-Gal4, and ii), we added repressible operators within the array, which are targeted exclusively by mutTetR-Mxi1. The

operators upstream are in fact next to the array at the core promoter, rather than the UAS. Therefore, only one location is required for effective silencing of the array, and that is directly surrounding gRNAs within the array. We have now clarified this in the text (**Line 114-116**).

Line 114-116: “We specifically target the mutTetR-Mxi1 protein to surround clusters of gRNAs to silence transcription across the entire array in the absence of inducer, without recruiting rtTA-Gal4 to these sites and interfering with array transcription, with the upstream sites targeted to a core promoter adapted from Chen et al³⁰ (Fig. 1c, Design 3).”

2. In multiple places (e.g. lines 107, 213), the authors state the discovery that gRNA arrays can be transcribed without a promoter and propose that the promoter-matching guides are driving transcription. I found this claim dubious and in need of supporting data to stand. For one, the introduced terminator may be leaky (especially when combined with upstream transcription in the plasmid) or could include a cryptic promoter, both of which would mean the selected spacers have nothing to do with leakage. This could be explored using encoded reporters or RNA quantification (e.g. RT-qPCR, RNA-seq) of the transcribed array.

We agree that we had not ruled out upstream read through or cryptic expression when using the *CYC1* terminator to insulate the array in the No promoter condition to fully support this claim. As per your suggestion, we designed a new experiment where we substituted all the gRNA arrays shown in Fig. 1 with the GFP CDS (plus a strong Kozak sequence) and reassessed basal transcription using fluorescence (Supplementary Fig. 1). This revealed no detectable expression of GFP when using the *CYC1* terminator upstream of GFP, confirming readthrough from the neighbouring gene is not present and the *CYC1* terminator does not seem to act as a cryptic promoter in this instance, supporting our previous statement at **Line 101-102**.

Line 101-102: “This led us to the key discovery that gRNA arrays can transcribe without a promoter (Fig. 1c+d, No promoter; Supplementary Fig. 1).”

3. One of the central arguments is that using an inducible gRNA array helped stabilize the CRISPRai construct and minimize any growth defects. There are a few aspects that need to be addressed and improved. First, the growth measurements in Figure 2D need statistical analyses and would be stronger if the experiment was conducted as a competition (e.g. mixing barcoded strains with the different constructs). Second, the demonstration of succinate production would be more compelling if the authors could show that one of the early designs (or constitutive gRNA array expression) is less stable or exhibits slower growth than the optimal design—specifically under uninduced conditions.

We have now added statistics to Fig. 2d to highlight promoter conditions which present a significant difference between the lowest expression of all components. We selected the promoter combination which demonstrated no significant difference in growth compared to the lowest expression (least burden), but still led to effective activation and repression (good CRISPRai performance). Additionally, we have updated the figure to make it clearer which promoter combinations we selected. The text has been modified to clarify our decision for promoter choices (**Line 144-149**).

Line 144-149: “Based on these findings, we chose to build the inducible CRISPRai toolkit with the weak *REV1*, *PSP2*, and medium strength *HTB2* promoters driving the expression of dCas12a-VP, dCas9-Mxi1, and *Csy4*, respectively, as higher expression did not incur a large performance benefit (no change to maximum repression and > 90 % of maximum activation) but did lead to a significant fitness cost. As rtTA-Gal4 and mutTetR-Mxi1 were already under the control of the weak *RAD27* and *POP6* promoters, we kept these fixed.”

To demonstrate the benefits of our inducible CRISPRai platform, as per your suggestion, we have compared the growth rate of an earlier design (Constitutive) to the inducible system (Design 3) in the uninduced condition (Supplementary Fig. 2d). The inducible system showed a clear benefit over the constitutive system, where a significant reduction in growth was seen. Alongside the differences between Constitutive and Design 3 during transformation (Supplementary Fig. 2c), the inducible system has clear benefits for strain handling in the uninduced state.

We assess the stability of inducible and constitutive arrays in Supplementary Fig. 6, where we compare early induction to late induction in serial passaged cultures. The uninduced array was stable over at least 6 days of daily passaging, whereas the cultures induced at 0 h (proxy for constitutive expression) started to fail after 2 days of daily passaging. In the revised manuscript we now comment to the reader that our system is best suited to batch culture experiments, where we have demonstrated stability over at least 5 days for culturing. Notably this is an appropriate time- scale for almost all yeast metabolic engineering experiments.

Other comments:

4. Line 31: Could you use “coordinated” rather than “simultaneous”? “simultaneous” can be understood to mean that one gene is being both activated and repressed.

We thank the reviewer for this suggestion. This is a much clearer descriptor for CRISPRai in this instance. We have amended the text at this location and first line of the discussion (line 31 and 195)

5. Line 53: constraint rather than constrain

Corrected.

6. Line 82: specify that you’re using strains of yeast.

Thank you for pointing this out. This has also been amended in results (Line 153).

7. Introduction: I recommend citing PMID = 31270316 as an original example of combining Cas9 and Cas12a gRNAs in the same array.

Thank you, this is a nice example for *E. coli*, and have been included as an example (Line 55).

8. Line 91 – 93: More accurate phrasing would be helpful here, as the entire system isn’t regulated (CRISPRai is but not the Cas proteins), and the levels of the Cas proteins (versus the target genes) can be tuned.

We agree this was not well explained. We have amended the statement (**Line 87-88**). We hope this is now clearer.

Line 87-89: “In this way, CRISPR-based gene activation and inhibition can be regulated through the expression of a single transcript, and Cas protein expression can be tuned to balance CRISPRai performance with fitness.”

9. Line 106: in comparison to a no-gRNA control.

Added.

10. Line 118: I recommend commenting on the mechanism of repression so it's clear how Mxi1 would impact transcription not only downstream but also upstream of the operator site.

We have changed the description of Mxi1 as a “chromatin remodelling repression domain” to better explain what the mechanism of transcriptional repression (Line 112)

11. Line 200 – 206: the uninduced condition is brought up two separate places in the paragraph. Can the logic be slightly restructured so the uninduced condition is addressed only once?

We agree this paragraph was difficult to understand. We have rewritten this paragraph to simplify our comparison of all the different conditions, now using statistical measures (**Line 184-193**). We hope this is easier to follow.

Line 184-193: “We transformed the arrays into wildtype BY4741 yeast alongside a no-CRISPR control (WT), with the remaining auxotrophic markers introduced on a single-copy plasmid to create fully complemented strains for growth in minimal media³⁷. RT-qPCR confirmed we were indeed regulating the 11 genes in the intended manner, albeit to varying extents (Fig. 3e). In the induced state, a 45-fold increase in succinic acid production was seen in the Targeted strain over the WT strain after 2 days in batch culture (WT (induced) = 9.38 ± 5.7 mg/L, Targeted (induced) = 426.9 ± 13.3 mg/L), representing a 16-fold change in succinic acid when compared to the uninduced Targeted strain (26.4 ± 0.5 mg/L) (Fig. 3f). Finally, no significant difference in succinic acid titres were measured between all conditions excluding the induced Targeted strain, demonstrating that the increase in succinic acid was exclusively caused by CRISPRai and inducibility is highly controlled, as seen in our previous experiments with the regulation of fluorescent protein expression.”

12. Figure 2F: can you introduce gray ends on the primers to exactly match the resulting dsDNA fragment?

Added.

13. Figure 3B: I recommend adding a colored dot with each vertical axis so it's more obvious which line goes with which axis.

Thank you for the suggestion. We have coloured the fluorescent protein names in the axis titles to make it easier to identify which data corresponds to which axis.

14. Suppl. Figure 3: The bottom segment of each plot appears to be missing a label.

Fixed.

Reviewer #2 (Remarks to the Author):

The manuscript presents a CRISPR/Cas-based approach to bi-directional transcriptional regulation in yeast with inducibility of large gRNA arrays. The presented work is scientifically sound and demonstrates 8- and 9-fold bi-directional transcriptional regulation of at least two distinct genes simultaneously. The system leakiness was reduced with design 3 while maintaining high inducibility. Tuning of Cas protein expression was investigated for reducing metabolic burden, and a clear design- and cloning strategy for constructing large gRNA arrays is included. The main novelty that the authors claim is first use of simultaneous transcriptional activation and repression in combination with large gRNA arrays and inducibility.

Simultaneous transcriptional activation and repression is established literature, and so is multi-gene regulation with large gRNA arrays and inducibility of such. While I do appreciate the efforts to reduce system leakiness, metabolic burden to the cell, and making the system easy to clone and use in yeast, I find that the novelty does not merit publication in the current journal.

Thank you for your comments and we agree simultaneous transcriptional activation and repression is well established. However, to date, no one system has yet to combine simultaneous activation and repression, large multiplexing capacity, and inducibility. For multiplexed CRISPRai applications, the combination of all these attributes is highly advantageous, and in some instances necessary, such targeting genes that cause unknown synthetic lethality. For example, the engineering of metabolic pathways often requires the manipulation of many genes. As the number of targets increase, so do the chances of growth defects or genetic circuit failure, which can only be addressed with inducibility. Indeed, this was the core motivation for our work, as we experienced large growth defects during our earlier work when only targeting 3 promoters, all of which are non-essential (repurposed used as our model system in Fig. 1 to develop inducible arrays).

The discovery that gRNA arrays can in fact express themselves in the absence of a promoter, while seemingly obvious in retrospect (as arrays contain many 20 bp fragments of the promoters they are targeting), has not been reported and was key to understanding how to reduce system leakiness. This required developing several systems to finally untangle this effect and establish a novel system for silencing the entire array in the absence of the inducer, while allowing many gRNAs to be delivered. Furthermore, the design of the inducible gRNA arrays developed here rely on commonly used proteins and transcribe from a pol II promoter, and so should be transferable to other species. CRISPRai holds a lot of promise for rewiring transcriptional networks, and as shown here can be used to greatly improve the production of a metabolite in a single transformation. We hope these contributions make it much easier for scientists to achieve this in yeast, with principles that can be applied to other areas of biology.

Major recommendations:

1. I suggest that you use ANOVA for statistical analyses with Prism (GraphPad) where relevant.

We appreciate this suggestion. We have now performed ANOVA to all data where appropriate, which have improved our work. We have updated the manuscript and figures to reflect these analyses.

2. With regards to Fig. 3d+e you introduce 11 gRNAs targeting central metabolism for repression and activation to achieve 45-fold improvement of succinic acid titers relative to wild-type. This strategy shows that in situ design and introduction of multiple gRNAs is applicable to improving cell factories, without testing their individual activity first. However, the output (succinic acid) does not give information on how many promoters were regulated during system induction and to what extent.

This is a problem mainly because the manuscript, in my opinion, demonstrates bi-directional transcriptional regulation with certainty of no more than two genes simultaneously at this point.

Further, the fluorescence analysis on expressing multiple gRNAs in Supplementary Fig. 1b is unable to give a complete insight into expression and activity of individual gRNAs when they are expressed in combination. In fact, repression of mTagBFP2 appears to be slightly relieved with 7 or 8 gRNAs targeting the HHF1 promoter being co-expressed. I miss clear

evidence that all gRNAs in a 24-gRNA array are indeed functionally expressed, and to what extent. For example, Zhang et al. 2019 (<https://doi.org/10.1038/s41467-019-09005-3>) noticed that expressing >5 gRNAs in array in *S. cerevisiae* resulted in decreased frequency of gene disruption, and they speculate that low gRNA expression is the underlying cause (granted, a different gRNA expression strategy was applied).

At the very least, I suggest making a comparative analysis clearly showing the transcript level for all 11 target genes in targeted and untargeted strains used for quantifying succinic acid titers.

Alternatively, and even better for claiming full gRNA array activity, prior to showing Fig. 3d+e, you may consider targeting 24 different promoters simultaneously for activation and repression and analyze transcript levels of these genes relative to an untargeted control. As you perform your analysis, pay attention to both activity and position for each gRNA in the array. Is there a trend here that may influence how you and others will design large gRNA arrays in future?

We agree the fluorescent protein study was not sufficient to show large multiplexing capacity of our system, and we apologise for not including RT-qPCR data for the 11 genes we targeted for. Unfortunately, it was very hard to get reagents and access to a light cycler last year due to COVID.

Following your request, we have now added RT-qPCR data for the 11 genes in the presence and absence of the inducer and show that they do indeed behave as expected. These results show that at least 11 promoters can be targeted using our system. We agree that the 3-promoter study shown in Supplementary Fig. 2 that contains up to 24 gRNAs does not confirm their individual expression, although this experiment can say that up to at least the 18th gRNA in the 8x array is being expressed, due to the location of the gRNAs in the array that are targeting the *HHF1* promoter (driving mTagBFP2) and matching repression of mTagBFP2 seen in the shorter arrays.

We noticed that it is common for multiplex CRISPR papers to state the number of gRNAs that are delivered on an array without individually confirming expression by direct or indirect methods, as seen in this recent Nature Methods where they claim 25 gRNAs but show no more than 15 as promoters are targeted more than once (<https://doi.org/10.1038/s41592-019-0508-6>). To avoid confusing interpretations of our work, we have readdressed the manuscript to make it clear that the 24 gRNAs are delivered in the array, which we have shown can be silenced in the absence of inducer. We have also stated that we were able to regulate 11 genes, which we have now been confirmed by RT-qPCR. We have removed or reworded all sentences that could indicate that 24 genes were regulated. For example, one sentence from the discussion was reworded. It originally read: “We incorporated this new method into a highly tuned and easy-to-use CRISPRai toolkit for inducible up- and down-regulation of up to 24 genes in the industrially relevant yeast, *Saccharomyces cerevisiae*”. This has been changed to: “We incorporated this new method into a highly tuned and easy-to-use CRISPRai toolkit that can deliver up to 24 gRNAs on a single array for inducible up- and down-regulation of genes in the industrially relevant yeast, *S. cerevisiae*”.

We also appreciate the suggestion of characterising how the position of the gRNA in the array may affect activity. While this is out of the scope of the current work, it is something we would love to conduct in a future study to help identify design rules for CRISPRai in yeast.

As a summary, as suggested by the reviewer, we have used RT-qPCR to prove that our CRISPRai toolkit was able to regulate 11 target genes to varying extents which lead to a 45-fold increase in succinic acid production. Our fluorescence assay also indicates that at least up to 18 gRNAs would be active. To our knowledge, these numbers are the largest

described in multiplexed engineering the widely used yeast *S. cerevisiae* and the largest in any organism demonstrating inducible and simultaneous activation and repression of target genes, which open the venue to a myriad of works in metabolic engineering and other fields.

3. Can you give some consideration in your discussion to the persistent leakiness of the system (comparing titers presented in line 204) in relation to metabolic burden and possibly how to further avoid it?

After doing a 2 way ANOVA, there is no significant difference between WT and Targeted uninduced systems, suggesting that the our CRISPRai system is highly controlled. However, looking at the values, there is a notable difference in the uninduced state (WT uninduced = 13.9 ± 3.0 mg/L and Targeted uninduced = 26.4 ± 0.5 mg/L), although this can be partially attributed to the presence of the CRISPRai system itself (Untargeted uninduced = 19.3 ± 0.6 mg/L). Even at the lowest expression, there is still noticeable burden from the CRISPR proteins (Fig. 2d). Following the advice of the reviewer, we have added a discussion of the burden of the CRISPRai system and how we might further avoid it in future systems by assessing alternative Cas proteins (**Line 209-212**). Additionally, we recommend limiting sub-arrays to a maximum of 6 gRNAs to ensure efficient silencing as the best way to avoid persistent leakiness (**Line 160-162**).

Line 160-162: "The limit of 6 gRNAs per vector or sub-array (24 gRNAs when sub-arrays are added together) is recommended to ensure a tight off state by keeping the distribution of mutTetO sites within the limits of mutTetR-Mxi1 silencing, and additionally simplifies validation of array identity by Sanger sequencing."

Line 209-212: "Tuning the expression of the CRISPR proteins allowed us to balance performance of the system with metabolic burden. However, a penalty for expressing Cas9-Mxi1, dCas12a-VP, and Csy4 still remains. Future work to determine alternative Cas proteins which impose a reduced fitness cost could be used to further improve the system."

Minor recommendations:

1. Cite Qi et al. 2013 for coining the term CRISPRi and for dCas9. doi: 10.1016/j.cell.2013.02.022.

Added at line 39: "In the simplest form, CRISPRai systems are achieved using a single catalytically inactive Cas protein⁵,"

2. Line 35: Insert citation(s) (e.g., Zalatan et al. 2015 doi: 10.1016/j.cell.2014.11.052)

Reference added (Line 30).

3. Line 73: Introduce abbreviation for *Saccharomyces cerevisiae* (*S. cerevisiae*) and use throughout.

Done.

4. Fig. 2b: I would like the promoter-codes nuances to be clearer/more distinct.

We have added a more distinct legend for the promoter codes and additionally highlighted the promoter combinations that were selected.

5. Fig. 2d: Do growth rates differ significantly?

We have performed an ANOVA to determine any significant difference to the lowest strength promoter combinations (lowest burden possible). We have also modified the text to describe the differences and our motivations for choosing the promoters combinations (**Line 144-148**).

Line 144-148: “Based on these findings, we chose to build the inducible CRISPRai toolkit with the weak *REV1*, *PSP2*, and medium strength *HTB2* promoters driving the expression of dCas12a-VP, dCas9-Mxi1, and Csy4, respectively, as higher expression did not incur a large performance benefit (no change to maximum repression and > 90 % of maximum activation) but did lead to a significant fitness cost.”

6. Line 200-201: “Fig. 3f” should be corrected to Fig. 3e.

The addition a Fig. 3e has It is now corrected the error.

7. Line 203-204: “No major differences”. Replace with statistic measures.

A 2wayANOVA multiple comparisons was performed between the WT, Untargeted, and Targeted conditions in the presence and absence of inducer, and no statistical significance was determined between the uninduced conditions. We have updated the final paragraph to include these statistical measures and improved the wording (**Line 190-193**).

Line 190-193: “Finally, no significant difference in succinic acid titres were measured between all conditions excluding the induced Targeted strain, demonstrating that the increase in succinic acid was exclusively caused by CRISPRai and inducibility is highly controlled, as seen in our previous experiments with the regulation of fluorescent protein expression.”

8. Line 342: Use subscript for “600” in “OD600”.

Corrected.

9. Line 325: Insert citation for BY4741.

Added.

Reviewers' Comments:

Reviewer #1:

Remarks to the Author:

The authors have taken steps to address all comments raised by the reviewers, although some gaps and unsupported claims remain.

Reviewer 1, Comment 1: it still remains unclear what impact the two changes made to generate design 3 had. The authors posit that one site is likely more important than the other, although the provided quote was already present in the original manuscript and does not help settle this question.

Reviewer 1, Comment 2: even with the insertion of a GFP reporter, there is still a logical jump from "no measurable GFP expression with the upstream terminator" to "there's transcription from the CRISPR array." Direct evidence of transcription would be needed yet remains absent. In this case, I think it's fine if the authors make a padded claim that the data support such a mechanism, although nothing definitive can be said.

Reviewer #2:

Remarks to the Author:

I appreciate all updates. The work is scientifically sound and will indeed become useful for e.g., metabolic engineering purposes, and I no longer oppose publication.

I do have a few last recommendations that will help you conform with the journal's guidelines before publishing:

- Make statistical analysis for Fig. 1A+B&D and Fig. 3E and insert asterisks.
- Explain what p-value **** refers to (Fig. 3F) in figure legend. This should be done in all figures and legends.
- Consider showing individual data points / replicates on all graphs.

Sincerely,

Emil

Reviewer #1 (Remarks to the Author):

The authors have taken steps to address all comments raised by the reviewers, although some gaps and unsupported claims remain.

Reviewer 1, Comment 1: it still remains unclear what impact the two changes made to generate design 3 had. The authors posit that one site is likely more important than the other, although the provided quote was already present in the original manuscript and does not help settle this question.

We thank reviewer 1 for bringing this up again and help us improve the readability of our manuscript. To clarify, the single design change we wanted to make from design 2 to design 3 was to target repression to the array in the uninduced state. However, to target the array without interfering with transcription requires two orthogonal Tet systems, otherwise the Tet-ON system would also target the array in the on state. To achieve this, we therefore had to make two changes: i) removing repression from the UAS, and ii) adding repression to the surround the gRNAs. Unfortunately, there is no intermediate design that can be tested to answer this question as the two Tet systems would conflict.

We have now revised the first sentence of the paragraph to outline our motivations for the design and clarified the need for orthogonality that shaped these design changes in the final sentence. The paragraph now reads as follows:

“To solve this problem, we redesigned the system to now focus on silencing the array instead of the upstream promoter in the uninduced state using the opposing actions of an orthogonal Tet-ON and Tet-OFF system. (**Fig. 1c, Design 3**). The Tet-ON system is composed of the reverse TetR protein fused to the Gal4 transcriptional activation domain (rtTA-Gal4)²⁸. This protein binds to Tet operator (TetO) sites upstream of the 5' UTR in the presence of inducer to drive expression of the gRNA array. The Tet-OFF system uses a mutated version of the TetR protein (E37A P39K) fused to the chromatin remodelling repression domain Mxi1 (mutTetR-Mxi1), and binds to an orthogonal TetO variant sequence (Tet4C5G, mutTetO)²⁹. We specifically target the mutTetR-Mxi1 protein to surround clusters of gRNAs to silence transcription across the entire array in the absence of inducer, without recruiting rtTA-Gal4 to these sites and interfering with array transcription, with the upstream sites targeted to a core promoter adapted from Chen et al³⁰.”

Reviewer 1, Comment 2: even with the insertion of a GFP reporter, there is still a logical jump from “no measurable GFP expression with the upstream terminator” to “there’s transcription from the CRISPR array.” Direct evidence of transcription would be needed yet remains absent. In this case, I think it’s fine if the authors make a padded claim that the data support such a mechanism, although nothing definitive can be said.

We agree with reviewer 1 that no GFP expression does not necessarily mean zero transcription although as reviewer 1 also suggests, these data support the proposed mechanism while more experiments would be required to fully confirm it.

As suggested by reviewer 1, we have now included a sentence to clarify that additional experiments would be required to further characterise the mechanisms by which CRISPR arrays are expressed in the absence of promoter.

The added sentence reads as follows: "Since gRNA arrays that target promoters are themselves made of 20 bp fragments of those promoters, we reasoned that these short sequences are sufficient to clear nucleosomes. This may allow the transcriptional machinery to gain access and initiate transcription from within the array, although further investigation is required to characterise the exact mechanisms by which expression of the gRNA array occurs."

Reviewer #2 (Remarks to the Author):

I appreciate all updates. The work is scientifically sound and will indeed become useful for e.g., metabolic engineering purposes, and I no longer oppose publication.

I do have a few last recommendations that will help you conform with the journal's guidelines before publishing:

- Make statistical analysis for Fig. 1A+B&D and Fig. 3E and insert asterisks.
- Explain what p-value **** refers to (Fig. 3F) in figure legend. This should be done in all figures and legends.
- Consider showing individual data points / replicates on all graphs.

We thank the reviewer for their positive feedback. We have amended the manuscript with the suggested changes.